# Strength and Durability of Sustainable Self-Consolidating Concrete with High Levels of Supplementary Cementitious Materials

**DOI:** 10.3390/ma15227991

**Published:** 2022-11-11

**Authors:** Moslih Amer Salih, Shamil Kamil Ahmed, Shaymaa Alsafi, Mohd Mustafa Al Bakri Abullah, Ramadhansyah Putra Jaya, Shayfull Zamree Abd Rahim, Ikmal Hakem Aziz, I Nyoman Arya Thanaya

**Affiliations:** 1Department of Surveying Techniques, Technical Institute of Babylon, Al-Furat Al-Awsat Technical University (ATU), Najaf 54003, Iraq; 2Tech Remix LLC, CGRC+Q8C, Jiddah St, Al Jerf Industrial 1, Ajman P.O. Box 4778, United Arab Emirates; 3Department of Water Resources, Faculty of Engineering, Al-Mustansiriyah University, Baghdad 10052, Iraq; 4Faculty of Chemical Engineering & Technology, Universiti Malaysia Perlis (UniMAP), Arau 02600, Malaysia; 5Centre of Excellence Geopolymer & Green Technology (CEGeoGTech), Universiti Malaysia Perlis (UniMAP), Perlis 01000, Malaysia; 6Faculty of Civil Engineering Technology, Universiti Malaysia Pahang, Lebuhraya Tun Razak, Kuantan 26300, Malaysia; 7Department of Civil Engineering, Udayana University, Bali 80361, Indonesia

**Keywords:** self-consolidating concrete, SCC, fly ash, GGBS, microsilica, sustainable concrete, high strength, durability

## Abstract

Self-consolidating concrete (SCC) has been used extensively in the construction industry because of its advanced characteristics of a highly flowable mixture and the ability to be consolidated under its own weight. One of the main challenges is the high content of OPC used in the production process. This research focuses on developing sustainable, high-strength self-consolidating concrete (SCC) by incorporating high levels of supplementary cementitious materials. The overarching purpose of this study is to replace OPC partially by up to 71% by using fly ash, GGBS, and microsilica to produce high-strength and durable SCC. Two groups of mixtures were designed to replace OPC. The first group contained 14%, 23.4%, and 32.77% fly ash and 6.4% microsilica. The second group contained 32.77%, 46.81%, and 65.5% GGBS and 6.4% microsilica. The fresh properties were investigated using the slump, V-funnel, L-box, and J-ring tests. The hardened properties were assessed using a compressive strength test, while water permeability, water absorption, and rapid chloride penetration tests were used to evaluate the durability. The innovation of this experimental work was introducing SCC with an unconventional mixture that can achieve highly durable and high-strength concrete. The results showed the feasibility of SCC by incorporating high volumes of fly ash and GGBS without compromising compressive strength and durability.

## 1. Introduction

Self-consolidating concrete (SCC) was invented in 1980 as a promising solution to cast concrete for structures with dense reinforced formwork sections [1,2]. Technically, SCC can be placed and consolidated in congested reinforced sections under its weight and flow around reinforcement by improving filling capacity. The cohesiveness of the concrete obtained from optimized mixture design and proper handling of concrete during pouring facilitates casting concrete without segregation and bleeding [1,3,4,5,6,7]. In addition, the characteristics of SCC provide more technical solutions by eliminating vibrating equipment, reducing noise pollution, and lowering labor costs and construction time. Applying SCC in the construction industry showed more positive aspects, such as reduced labor associated with lower human risk in construction sites. From the microstructural point of view, the proper mixture design of the SCC improves the interfacial transition zone (ITZ) between aggregate, reinforcement, and bulk cement paste, which enhances durability by decreasing permeability [2,7,8,9]. Given all the advantages of SCC in practice, emphasis has been placed on optimizing its constituent composition by incorporating supplementary cementitious materials (SCMs). Moreover, emphasis has been placed on investigating the effect of the water-to-binder ratio in the mixture design with the use of chemical admixtures such as superplasticizers and viscosity-modifying admixtures. SCMs and mixture design with the type of applications collectively affect SCC’s fresh and hardened properties [10,11,12]. However, with the increasing global trend towards sustainable development in construction, more research is indispensable to reduce the high content of Portland cement used in SCC production. SCC cost was considered one of the major drawbacks in the production process due to the high content of the OPC utilized in the mixture [13]. It has been estimated that cement production was 4100 million tons [14]. The production process of Portland cement releases at least 930 kg/ton of carbon dioxide into the atmosphere [15], which is considered one of the main challenges that many countries have targeted by adopting long-term measures to minimize CO_2_ emissions [16]. Furthermore, consuming natural resources for the constituent components of concrete exerts a considerable impact annually that jeopardizes sustainability. Due to the increasing population worldwide and rapid urbanization, there is global demand for Portland cement, which augments a massive demand in the construction industry for infrastructural development [17]. Therefore, inspecting more sustainable and environmentally friendly construction materials is crucial in developing advanced concrete-tech binders. Developing green concrete by incorporating SCMs such as fly ash, GGBS, and microsilica is a promising solution for producing environmentally friendly concrete by reducing mixtures’ OPC quantity and lowering CO_2_ emissions [18,19,20,21,22]. The SCMs have long-lasting effects on the environment because of their nature as non-biodegradable waste materials. Incorporating pozzolanic materials can improve concrete durability and increase the life span of the structures by reducing the required maintenance and repair in addition to cement reduction [23,24]. The most commonly used SCMs for replacing OPC in SCC are ground granulated blast furnace slag (GGBS), fly ash (FA), silica fume (SF), and Microsilica (MS) [25,26,27]. Fly ash, GGBS, and microsilica are by-products generated from different manufacturing processes and are not produced intentionally. The SCMs have been used as essential constituents to enhance concrete performance and durability when exposed to different aggressive environments [28,29]. Previously, fly ash, GGBS, and microsilica have been applied to replace OPC partially in SCC to enhance fresh and hardened properties and reduce its carbon footprint owing to the high content of binder used in its mixture design [30,31,32,33]. This technique was intended to lower CO_2_ emissions associated with OPC production and improve concrete durability [34,35,36]. Moreover, it is intended as a method to enhance the environment by applying a green combined binder with sustainability in addition to the durability factor [37]. However, increasing the cement replacement level while maintaining the engineering properties and the durability of SCC is still challenging [38]. Previously, it has been found that 10% silica fume and 10% GGBS gave the best results for the durability and mechanical properties in SCC; however, the recommendations were that 6% silica fume and 8% GGBS should be incorporated as a partial replacement for the OPC separately for better performance [39]. A silica fume to OPC ratio was used in three different percentages (4.85%, 10.5%, and 14%) to produce SCC; however, a better mechanical performance was exhibited in comparison to the normal vibrated concrete [40]. Zhao et al. [41] incorporated 20–40% FA as a partial replacement for OPC to investigate its performance in SCC and concluded that a decrease in mechanical properties was registered at 7 and 90 days in both mechanical properties. In another study, Liu [42] investigated the substitution of FA as a partial replacement for cement to study its effect on SCC. The research showed a decrease in compressive strength as FA increased from 20% to 80%. The results showed that 40% replacement with FA revealed insignificant compressive strength loss. Siddique’s results [43] also showed that up to 35% replacement with FA in SCC resulted in compressive strength reduction as well as split tensile strength. Replacing cement with FA was investigated by Uysal and Sumer [44]; they concluded that FA up to 25% may result in more developed compressive strength compared to the 100% OPC SCC. Previous studies investigated the influence of silica fume and fly ash on the performance of the SCC. Cement was replaced with 10% SF and 30% FA, and the results showed an improvement in the compressive strength [45]. The impact of GGBS and SF on SCC compressive strength was determined after replacing the cement with 30%, 50%, 65%, and 80% GGBS. SF was incorporated with 50% GGBS in three percentages, 5%, 10%, and 15%. The study showed that SF has a recognized impact on compressive strength when used with 50% GGBS [46]. In another study, SF was incorporated as a partial replacement for OPC up to 25%. The results showed an enhancement in tensile strength, whereas a decrease in compressive strength was registered; however, researchers concluded that no more than 5% SF may be used as an enhancement factor in SCC [47]. Micro- and nanosilica were investigated as replacements for OPC in high-performance SCC, and the results showed the dominancy of nanosilica in its effect on the strength properties due to its high reactivity. It is concluded that particle size distribution with a wider range may create low porosity and low water demand and enhance packing density [48]. Higher resistance of sorptivity characteristics of SCC was registered when combined with FA and SF; however, partial replacement of OPC with only 20% FA showed a reduction in sorptivity [49].

In general, fewer and limited studies have been performed regarding the durability of the SCC with the maximum amounts of binary blended replacement of OPC by SF and GGBS; moreover, fewer studies have been conducted to investigate the durability of SCC using SF and FA. In the present experimental investigation, this study investigates up to 70% cement replacement with binary mixtures of microsilica, fly ash, and GGBS yet aims to maintain the engineering properties of SCC for infrastructure applications. Essentially, the novelty in this work is the sustainable mixture design associated with high-strength and durable SCC with high content of SCMs as a partial replacement for cement. This research will achieve two significant goals: the first one is the sustainability of SCC as a high-strength building material, and the second one is the advanced durability which will provide protection against an aggressive environment. Two groups of SCCs were designed to study fresh properties such as flowability and viscosity, in addition to compressive strength as the hardened property. In order to evaluate SCC durability and service life [50], a water absorption test, water permeability test, and chloride ion penetration test were applied. The first group of mixtures contained a binary system with up to 38.74% low calcium fly ash having 0.12 CaO/SiO_2_ in addition to the microsilica. The second group contained GGBS at up to 71.16% having 1.33 CaO/SiO_2_ in addition to the microsilica. Microsilica was incorporated in a constant quantity of 30 kg/m^3^ in all mixtures, which is equal to 6.4%. The main objective of the current work is to confirm the possibility of producing high-strength and durable SCC by incorporating a high percentage of SCMs as a partial replacement for OPC.

## 2. Materials and Methods

This study used ordinary Portland cement with 42.5 N grade in compliance with BS EN 196 [51] and standard BS EN 197–1:2000 CEM I [52]. The chemical and physical properties of OPC are shown in Table 1 and Table 2. The supplementary cementitious materials were ground granulated blast furnace slag GGBS complies with the BS EN 15167–1:2006 [53]S, Indian low calcium fly ash (FA), and microsilica (MS). Microsilica (MS) was used in a constant quantity (30 kg/m^3^) in all mixtures. Table 3 and Table 4 show the chemical analysis for all SCMs and the residue on 45 micron sieve, respectively. Polycarboxylate high-range superplasticizer (HRSP) type F and G [54,55] compatible with the ASTM C494 and BSEN [55,56] was used to produce SCC. It is a high-performance concrete superplasticizer based on modified polycarboxylate ether, and it has a unique carboxylic ether polymer with long lateral chains. The superplasticizer, an effective cement dispersant and high-range water reducer, was used to fix constant water content and control flow in all mixtures. In addition to that, it can produce high-flowing concrete without segregation, high early strength, and high workability with lower water content and lower permeability. It is a high-range superplasticizer that can be used for ready-mix concrete, self-consolidating concrete, precast concrete, and underwater concreting. Moreover, it is used for concrete containing microsilica, GGBS, and fly ash with extremely low w/c [57]. Polycarboxylate high-range superplasticizer (HRSP) was used for the admixture in this research.

In order to overcome the problem of natural fine sand shortage, a mixture of fine washed sand and dune sand was used as part of the concrete ingredients in all mixtures. According to the sieve analysis, the dune sand particle size is 50% passing sieve size with 0.150 mm and 1% passing sieve size with 0.075 mm. Coarse aggregate was used in two sizes, 20 mm and 10 mm.

### 2.1. Experimental Program

The experimental program was designed to produce SCC with a high replacement level of OPC content by incorporating accurate amounts of several combinations of FA with MS and GGBS with MS. Table 5 shows the mixture proportions used in this experimental work. Seven SCC mixtures were prepared with a constant water-to-binder ratio (w/b) of 0.33. Different characteristics of SCC were investigated according to the ASTM [59] and European guidelines [60].

### 2.2. Testing Procedures

Fresh properties of SCC were determined by using the slump-flow test to determine the concrete flowability [59,61] (Figure 1) and V-funnel [62], L-box [63], and J-ring [64] tests, as shown in Figure 2, Figure 3 and Figure 4, respectively. SCC viscosity was assessed by measuring the flow rate using the V-funnel test. The L-box test was used to measure the passing ability of SCC [2], and the flow spread with passing ability was measured by using the J-ring test. The durability of SCC was measured by applying different tests that have been used regularly for standard concrete [20,21,22]. Water absorption (Figure 5) was determined according to BS 1881: 122 [65]. Water permeability (Figure 6) was determined according to BS EN 12390 [66]. The rapid chloride penetration test (RCPT) (Figure 7) was conducted for all concrete mixtures to measure the electrical conductance and ability to resist chloride ion penetration. The RCP test was conducted according to ASTM C 1202 [67]. Figure 8 shows sample preparation.

## 3. Results and Discussion

### 3.1. Workability

#### 3.1.1. Slump Flow

Figure 9 shows the initial slump flow and the slump retention results for the self-consolidating concrete mixtures. As can be seen, two periods were chosen to measure a range of flow of SCC under its weight, initially before casting and 60 min after casting.

In general, the initial slump and 60 min slump results of SCC mixtures with SCMs were increased in comparison to the reference SCC mixture that was produced with 100% OPC. The increase in flow is related to the high replacement levels of FA, GGBS, and MS. The enhancing effect of supplementary cementitious materials on the flowability of concrete was reported in previous studies [20,68]. This behavior is attributed to the positive effect of SCM particles because of their high surface area on the packing density of the mixtures and the lower reactivity of the SCMs compared to the OPC.

FA presented different effects on SCC initial flow compared to the effect of GGBS. As shown in Figure 9, FA with MS showed gentle concave initial flow and a decrease in the measurements. Replacement of OPC with 20.43%, 29.5%, and 38.74% FA and MS showed 740 mm, 730 mm, and 720 mm initial flow, respectively. On the other hand, the replacement of OPC with GGBS showed a sudden increase in the initial flow. As can be seen, 38.74%, 52.6%, and 71.16% GGBS and MS replacement showed 720, 750, and 750 mm initial flow, respectively. The effect of particle size and the large surface area that was added to the mixture effectively changed the behavior of the mixtures and the initial slump.

Two periods were applied to measure the slump in this investigation for different reasons: The first reason was the SCC workability and high-range superplasticizer (HRSP) dosage compatibility with ingredients having different particle sizes; moreover, the time tolerance for SCC to be handled and cast was considered. On the other hand, HRSP admixture was added to the mixture in order to keep the w/b ratio fixed at 0.33. The figure shows that the dosage was gradually increased with the increase in the replacement ratio of the cement [69].

Figure 10 shows the gradual increase in HRSP dosage with fly ash and GGBS mixtures. As can be seen, the admixture dosage was increased gradually, which may be attributed to the higher specific area of the cementitious materials [70]. Slump and workability showed that incorporating FA and GGBS in addition to a constant quantity of MS results in almost converging quantities of HRSP admixture needed to keep a constant water-to-binder ratio of 0.33. Moreover, it has been reported that MS may increase the water demand in the concrete mixture due to its very fine smooth spherical glassy particles that provide a high surface area compared to FA, GGBS, and OPC [71].

Generally, SCC produced with different amounts of supplementary cementitious materials has shown an acceptable range of slump and workability. Such flow ability may provide appropriate time for handling and casting the mixture for different applications and environments. According to the European guidelines for self-compacting concrete, the slump flow between 660 and 750 mm for SCC mixtures is suitable for many normal applications such as walls and columns [2,72]. It has been reported that FA and GGBS show a slow hydration reaction; however, providing sufficient moisture content will allow the reaction to be continued over a longer period of time. This mechanism will affect the concrete ability to flow and setting time; moreover, it will affect the strength development [73].

#### 3.1.2. V-Funnel Test

Figure 11 shows the V-funnel test results for the SCC mixtures produced with FA and GGBS. As can be seen, the incorporation of FA and GGBS at different replacement levels registered different rates of flow in the V-funnel test. Mixtures produced with FA and MS had a lower rate of flow which increased as the percentage of replacement increased in comparison to the reference mixture. The registered rate of flow was 8, 5, and 4 s for SCC having 20.43%, 29.5%, and 38.74% FA and MS, respectively. The incorporation of GGBS showed a different effect in comparison to the reference mixture. As can be seen, there was an increase in the rate of flow with the increase in GGBS percentage. The SCC mixture with 38.74%, 52.6%, and 71.16% GGBS and MS showed an increase in the rate of flow, which was 8, 10, and 12 s, respectively. Overall, the V-funnel test can provide an indication of SCC viscosity by measuring the time required for the mixture to pass the V-funnel. The concrete viscosity increases with the increase in flow time. The results showed that FA decreased the concrete viscosity while GGBS increased the SCC viscosity.

According to the European guidelines, SCC with low viscosity will present a very quick initial flow that will then stop, whereas SCC with high viscosity may continue to flow over an extended time (creep over) [60]. The results may reflect the ability of the produced mixtures to show adequate filling capability even with congested reinforcement and the capability for the mixture to be self-leveled with the best surface finish; however, it has been reported previously that SCC may suffer from bleeding and segregation [2,8].

In this investigation, based on visual observation during the V-funnel test, mixtures showed no bleeding and no segregation, which reflects an advanced design and performance. The rate of flow showed better times in all mixtures which are lower than 100% OPC-SCC. It is practical to mention that viscosity is also a critical parameter and is required to be measured for SCC where a good surface finish is in demand when reinforcement is very dense [8].

#### 3.1.3. L-Box Test

Figure 12 shows the passing ability (PA) ratio of all mixtures. For the OPC-SCC control mixture, the PA ratio was 0.85, which meets the requirements mentioned in the standards [60]. SCC produced by replacing OPC with FA or GGBS with the constant amount of MS showed a higher passing ratio. The SCC mixtures showed the ability to reach equal depths for vertical section height and a horizontal section height of the L-box container. According to the European guidelines for self-compacting concrete [60], the conformity criteria for L-box are classified into two classes based on the number of steel bars installed in the L-box. The first class is PA1 (two bars with 59 mm gap), and the second class is PA2 (three bars with 41 mm gap). This classification is related to the number of smooth steel bars (12 + 0.2 mm) installed at the gate of the filling hopper of the L-box used in this investigation. This test represents the ability of concrete to flow in spaces and pass through steel reinforcing bars or tight openings without aggregate segregation or blocking; moreover, it represents the ability of concrete to flow without leaving voids at the time of casting. European guidelines showed that the passing ability ratio should be ≥ 0.75, whereas British Standards (BSI) showed that the passing ability (PA) ratio for SCC must be ≥ 0.8 and should not exceed 1.0. Figure 12 shows that all the SCC mixtures had a passing ability equal to 1.0 except for SCC with 35% GGBS (PA = 0.9); however, the PA ratio of SCC with 35% GGBS was higher than the OPC-SCC passing ability ratio. In general, the results in this investigation showed that the SCC mixtures designed with replacement levels from 20.43% to 71.16% GGBS/FA have the ability to be self-leveled horizontally when placed in formwork. Moreover, cementitious materials have participated successfully in producing SCC with an appropriate passing ability, and no segregation or blocking was observed for the mixtures [2,63].

#### 3.1.4. J-Ring Test

Figure 13 shows the J-Ring test results expressed to the nearest 10 mm for all mixtures. In this experimental work, the J-ring test was used to assess SCC passing ability [60,74]. This is crucial to be sure that SCC can flow through spaces between and around congested reinforcement, through tight openings, and around any other obstructions that might prevent SCC from flowing during the casting process without segregation, blocking, or leaving voids. As can be seen from the figure, the spread flow ability of OPC-SCC is 600 mm, whereas all developed sustainable high-strength SCCs showed a higher ability to flow and spread within the casting process.

The results showed that there was an increase in the flow spread of concrete produced with binder containing FA and MS. The increase was 15%, 17%, and 17% for SCC produced by replacing OPC with 20.43%, 29.5%, and 38.74% of the FA + MS system, respectively. Replacing OPC with high levels of GGBS and MS also showed an increase in the spread flow ability. The results registered 12%, 15%, and 10% increases in the flow spread for mixtures produced with 38.74%, 52.6%, and 71.16% GGBS + MS, respectively. Generally, the test depicted the capability of SCC produced with high levels of replacement to fill the formwork without segregation or blocking even with congested reinforcement and the possibility of full compaction based on its weight.

### 3.2. Mechanical Performance

#### Compressive Strength

Figure 14 shows the compressive strength test results for SCC produced with 100% OPC and the binary binder system of FA with MS and GGBS with MS as a partial replacement for OPC. The test was conducted at 3, 7, and 28 days of curing, and the average of three specimens for each test was recorded. As can be seen from Figure 14, the OPC-SCC mixture had 62.5, 68.1, and 72.6 MPa at the ages of 3, 7, and 28 days, respectively. The results showed the ability of the mixture design to produce high-strength self-consolidated concrete at the early and late ages of 3 and 28 days, respectively. Most of the results showed an increase in strength with the incorporation of FA and GGBS. The increase in strength at the age of 28 days was 18%, 15%, and 10% for mixtures with 20.43%, 29.5%, and 38.74% FA and MS, respectively.

Furthermore, the increase in strength at the age of 28 days was 22%, 24%, and 13.4% for mixtures with 38.74%, 52.6%, and 71.16% GGBS with MS, respectively. It is observed that there was a slight reduction in strength for all mixtures with the increase in replacement levels at all ages; however, all mixtures showed high strength results in comparison to the reference SCC-OPC mixture. In general, the mixture design used in this experimental work showed the ability to produce high-strength concrete at early ages, meeting advanced concrete requirements. This may enable the de-molding of work forms and increase the constructability during the production cycle while maintaining a high sustainability index due to the high amount of replacement levels. The overall effect of cementitious materials was clear in increasing the strength property by replacing OPC in self-consolidating concrete [17,75]. It has been reported previously that a reduction in compressive strength property was registered for binary and ternary mixtures, and that was attributed to the low content of CaO which may cause a delay in hydraulic reaction [76], whereas, in this investigation, high-strength SCC was achieved.

The mixture proportions of the binder in this work may be a combination of synergistic ingredients that can chemically react well, producing a higher concentration of hydration products. On the other hand, the effect of the dune sand commingled with fine sand may have filled different size voids in the structure of the paste and aggregate, producing well-compacted concrete [21]. The results showed that GGBS was able to be used as an effective replacement material with good homogeneity and high synergy with MS to produce high-strength self-consolidating eco-friendly concrete despite the high level of replacement. The incorporated MS was effectively active during the chemical reactions with the presence of FA and GGBS, producing high-early-strength self-consolidating concrete. MS works as a booster to continue chemical reactions in the system, generating high-strength concrete at the age of 28 days. It is like a reactor that works to activate the potential chemical power in SCMs and react with Ca(OH)_2_ to form greater quantities of calcium silicate hydrate (C-S-H). This mechanism may work as a densification factor to fill different voids between paste ingredients and also fill small spaces between fine particles, thus enhancing the structure of the SCC by increasing the packing density and producing a denser microstructure. The synergy between FA, GGBS, and MS previously was reported in [77]. Moreover, the polycarboxylate high-range water superplasticizer used in this mix design was capable of accelerating and boosting the chemical reactions, increasing the hydration products.

The homogeneity of the mixed materials for binder production, crushed fine sand, dune sand, and coarse aggregate was also a vital factor in producing concrete with a density between 2460 and 2485 kg/m^3^, as presented in Table 6.

The ITZ in normal concrete, which is the space between the binder paste and the coarse aggregate particles, exhibits lower strength than bulk cement paste, which is attributed to the gathering of more voids. This weakness is due to the accumulation of bleed water underneath aggregate particles, resulting in difficulty in packing solid particles near the surface. This behavior leads to more calcium hydroxide (CH) forming and concentrating in this region than elsewhere.

In this investigation, 6.4% MS played a sophisticated role at an early age. MS increased the bond strength between the paste and aggregate particles. According to the ACI Committee 234R-06, MS will react with calcium hydroxide (CH), producing more calcium silicate hydrate (C-S-H), and it is expected that all CH will be consumed in the early ages, producing a well-crystallized form of CSH-I. Without the pozzolanic reaction of the added MS, CH crystals will grow large and tend to be strongly oriented parallel to the surface of the aggregate particles. CH is weaker than C-S-H, and when the crystals are large and strongly oriented parallel to the aggregate surface, they are easily cleaved. A weak transition zone results from the combination of high void content and large, strongly oriented CH crystals. Microsilica produces a denser structure in the transition zone with a consequent increase in microhardness and fracture toughness. The presence of MS as part of the binder in fresh concrete also may reduce bleeding and greater cohesiveness.

Moreover, different fine particles such as dune sand with MS may increase the packing of the solid materials as mentioned above. This behavior is related to the interlocking mechanism of the microparticles increasing the packing of solid materials by filling the spaces between cement and coarse aggregate grains [78]. It has been reported previously that GGBS can be used at an optimum level of up to 55% [70], whereas in this investigation, a high compressive strength was able to be produced with a higher replacement level of up to 71.16%.

### 3.3. Durability

#### 3.3.1. Water Permeability

Figure 15 shows the water permeability results for all samples exposed to a water pressure of 500 ± 50 kPa for a period of time extended up to 72 ± 2 h [66]. As can be seen from the figure, 100% OPC-SCC showed a 3 mm water penetration depth, whereas improved results showed water penetration resistance by FA and GGBS SCC. The SCC with 20.43% FA and MS replacement showed the same permeability as the reference concrete; however, there was a 67% reduction in water permeability when the replacement level increased to 29.5% FA and MS. The SCCs with 38.74% FA and MS and 38.74%, 52.6%, and 71.16% GGBS and MS showed zero water permeability. The results with zero water permeability can be related to the development of the hydration products with the homogeneous combination of the binder ingredients. The hydration products for FA with MS and GGBS with MS may be increased due to the pozzolanic reactivity acceleration. The permeability of concrete is a congregation of the size, shape, distribution, tortuosity, and continuity of the pores; overall, it is not a simple function. It has been reported previously that there is a good relation between concrete durability and maximum continuous pore radius [79]. In this investigation, it is suggested that different particle sizes for the fine sand with dune sand and large surface area for the particles of the cementitious materials in addition to the compactness of the hydration produced led to reduced pore size and cut off pore continuity. The mixture design mechanism priority targeted a great increase in the packing density by filling the micro- and nanopaste void–pore systems, which reduced the coefficient of permeability [80,81,82,83]. Moreover, the synergistic interaction of FA with OPC and MS or GGBS with OPC and MS may have refined the pore system generated in the cement gel that created and developed a very dense and complex structure inhibiting the penetration of water within the investigated duration of 72 h [84]. Improvement in concrete permeability may also be related to the superplasticizer effect, which is designed to lower concrete permeability, in addition to its different advantages with SCC.

#### 3.3.2. Water Absorption

Figure 16 shows the water absorption of SCC mixtures in percentage at 28 days. As can be seen, the water absorption results for all SCCs produced with SCMs showed lower results in comparison to the reference OPC-SCC. The water absorption test was performed according to BS 1881: Part 122 [65] and involved immersing specimens in water for 30 min after drying according to a certain procedure. It included calculating the increase in sample mass resulting from full water immersion and expressed as a percentage of the dried specimen. Replacing OPC with FA or GGBS and 30% MS showed a significant effect on enhancing the ability of SCC against absorbing water in a sophisticated way. The reduction in water absorption in mixtures with 20.43%, 29.5%, and 38.74% FA and MS was 33.3%, 40%, and 40%, respectively; mixtures with 38.74%, 52.6%, and 71.16% GGBS and MS showed 33.3%, 47%, and 53.3% reductions, respectively, as compared to the reference mixture. It has been reported previously that water may ingress to the surface of unsaturated concrete by capillary suction based on the initial water content [85,86,87]; moreover, capillary adsorption may be strongly connected to the size distribution of the pores in addition to pore volume and pore radius. Based on the work of Powers [88], two sizes of pores were identified; the smaller pores are the gel pores less than 10 nm in diameter working as part of the hydration products, and the larger pores are the capillary pores that occur due to excess water. In this investigation, the reduction in water absorption may be related to the synergistic interaction between supplementary cementitious materials. The hydration products of FA, GGBS, and water with different quantities of OPC in the presence of MS were developed and allowed the microsilica to react as any finely divided amorphous silica-rich constituent in the presence of CH. Calcium ions combined with the microsilica to form extra C-S-H through the pozzolanic reaction mechanism to produce a well-crystallized form of C-S-H type I which is formed during early age of curing [71,78]. It has been reported previously that fly ash and silica fume showed considerable a reduction in volume of large pores generated in concrete [89]. The same conclusion was reported when mixing silica fume and GGBS, which showed high early strength and later age strength development that may be related to the increase in the hydration products that reduced the pore volume size and structure in the mixture, resulting in reduced water absorption [90].

#### 3.3.3. Rapid Chloride Penetration Test (RCPT)

The RCPT was applied as quality control and to evaluate SCC chloride penetration. The evaluation included electrical conductance to provide a rapid induction of the chloride ion penetration resistance into the SCC. In this test method, according to ASTM standards, the amount of electrical current passed through 51 mm thick slices of 102 mm nominal diameter cores of cylinders for 6 h is monitored. Numerical results for the RCPT represent the total electric charge that can pass through the concrete [67]. It is important to mention that many factors affect chloride ion penetration, such as type of curing, w/b, the presence of polymeric admixtures, air-void system, aggregate type, degree of consolidation, and age of the sample when the test is applied.

As can be seen from Figure 17, the total charge passed through SCC produced with 100% ordinary Portland cement was 2700 coulombs, and this sample is classified as concrete with moderate chloride ion penetrability as reported previously [67]. In this investigation, SCC produced with FA and MS as well as GGBS and MS showed an advanced ability to reduce chloride ion penetrability effectively. Adding supplementary cementitious materials as a partial replacement for cement was extremely effective in producing SCC with very low chloride ion penetrability. All the charges passed through concrete samples had results between 170 and 340 coulombs, which are lower in an effective level than the result for the reference OPC-SCC having the same w/b.

Figure 17 shows that the reduction in the charges passed was 87%, 88%, and 90% compared to the reference OPC-SSC for SCC mixtures with 20.43%, 29.5%, and 38.74% FA and MS, respectively. The reduction in charge passed was 90%, 94%, and 94% compared to the reference OPC-SCC for SCC mixtures with 38.74%, 52.6%, and 71.16% GGBS and MS, respectively. Because of the diversity and non-homogeneous mixture of materials, the chloride ion penetration in concrete is a complex process of diffusion; moreover, other environmental factors are involved in the measurement (e.g., chloride ion concentration in seawater or structure location). Previously, it has been reported that the penetration process for chloride ions may be related to the pore system in the body of the concrete. The ions start the intrusion process into the pore system because of the diffusion process which will start due to the capillary suction [91]. The addition of SCMs has reduced the penetration of chloride ions efficiently and lowered ion diffusion ability to a very low level. This behavior may be attributed to the pozzolanic reaction resulting from the addition of SCMs which causes pore refinement. This process eventually reduced the concrete permeability, as shown in Figure 13 and Figure 14, which is also in agreement with the results of [82]. Combining or incorporating MS in the SCC mixture design as an activation factor was crucial to accelerate, enhance, and activate the chemical reactions with the presence of fly ash and GGBS. Microsilica worked as a reactor to activate the potential chemical power in SCMs and was an effective addition in increasing packing density, producing a denser microstructure [77].

## 4. Conclusions

In this experimental study, self-compacting concrete was produced with high replacement levels of OPC by SCMs. The results showed the ability to produce high-strength, highly durable concrete with a high quality of sustainability by reducing the OPC used in the SCC. Fly ash and GGBS were used as partial replacement materials with a constant quantity of microsilica. The following conclusions can be drawn based on the results registered from the experiments:In this investigation, GGBS and MS were able to be used at levels up to 71%. A sustainable and durable SCC was successfully produced.The high surface area of the SCM particles has increased both the initial slump and 60 min slump for the SCC in comparison to the 100% OPC-SCC. An appropriate time for handling and casting was registered. Moreover, SCC with high contents of SCMs showed the ability to be self-leveled with passing ability through reinforcement without segregation.Viscosity for SCC was decreased in mixtures containing fly ash, while with GGBS the viscosity was increased. The results showed adequate FA-SCC filling ability and low rate of flow with congested reinforcement, whereas GGBS increased the viscosity and consequently increased the SCC rate of flow. There was no bleeding and no segregation, which reflects an advanced mixture design. Flow spreadability was increased for SCCs with high levels of SCMs, which reflects the ability to flow and fill congested reinforcement formwork without segregation or blocking. SCC showed a higher passing ratio based on L-box test results.The synergy of high-content MS with FA or MS with GGBS was a clear factor in producing high-strength SCC. MS worked as a reactor to activate the potential chemical power in SCMs and react with Ca(OH)_2_ to form more calcium silicate hydrate (C-S-H).The combination of very fine SCMs in SCC showed an advanced interaction producing very dense cement gel with good compactness for the paste. The result showed sophisticated water permeability which is related to the effect of the increase in the hydration products and good compactness of different aggregate sizes and fine sand particles. Water permeability in SCC concrete was reduced to zero due to the effective changes in the gel pore system. Water penetration ability was reduced due to the final hydration products of high replacement levels of SCMs which reduced pore volume and changed pore structure.

## Figures and Tables

**Figure 1 materials-15-07991-f001:**
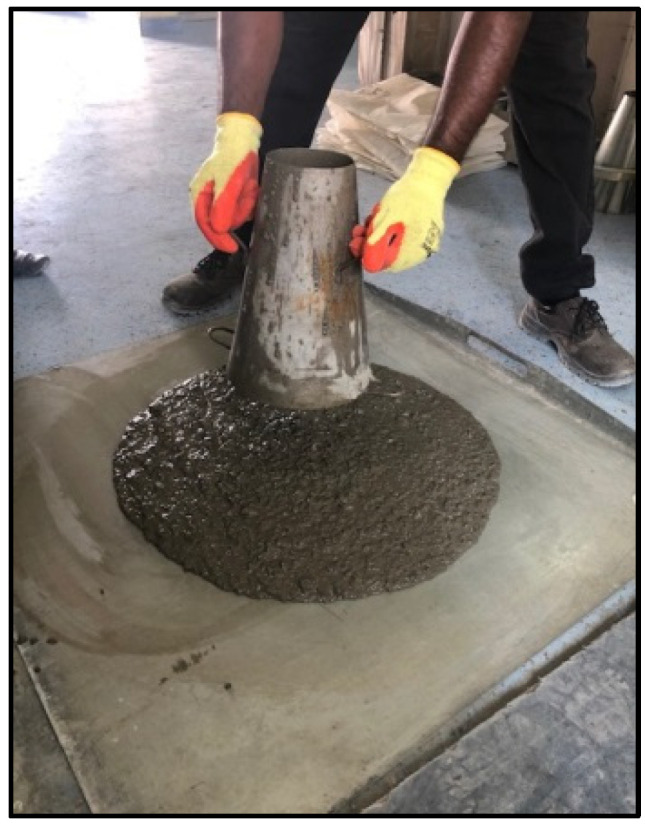
Slump-flow test.

**Figure 2 materials-15-07991-f002:**
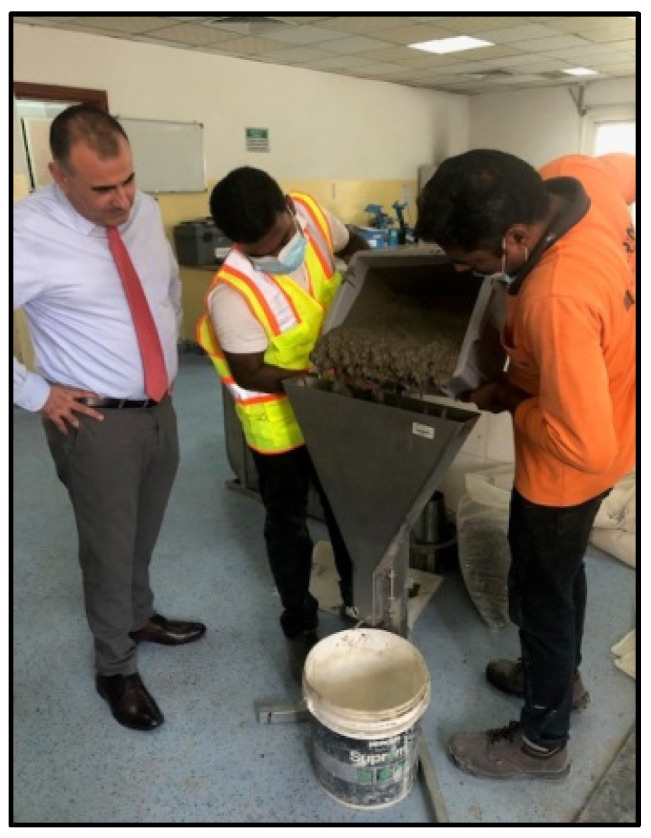
V-funnel test.

**Figure 3 materials-15-07991-f003:**
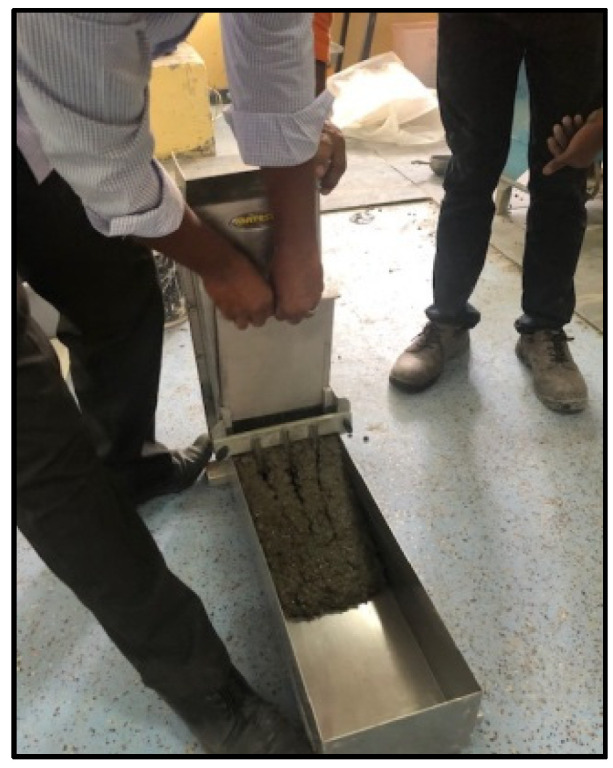
L-box test.

**Figure 4 materials-15-07991-f004:**
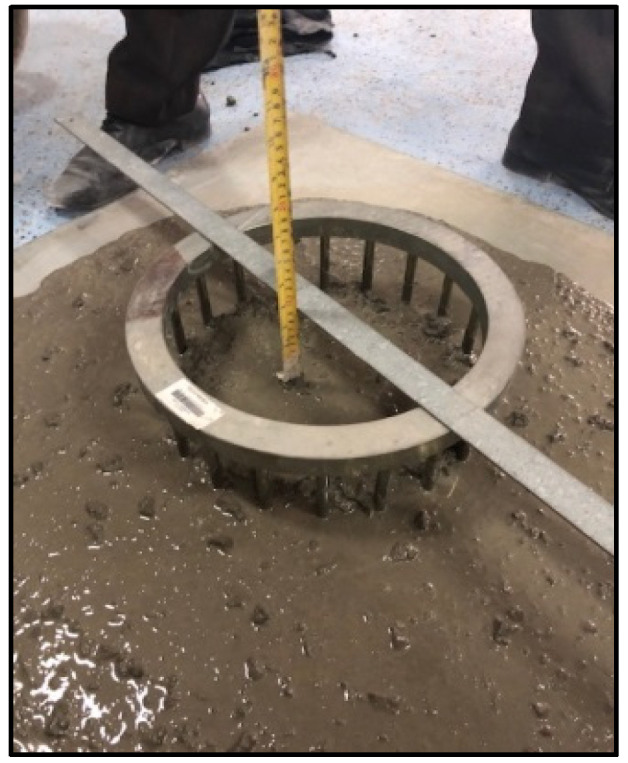
J-ring flow test.

**Figure 5 materials-15-07991-f005:**
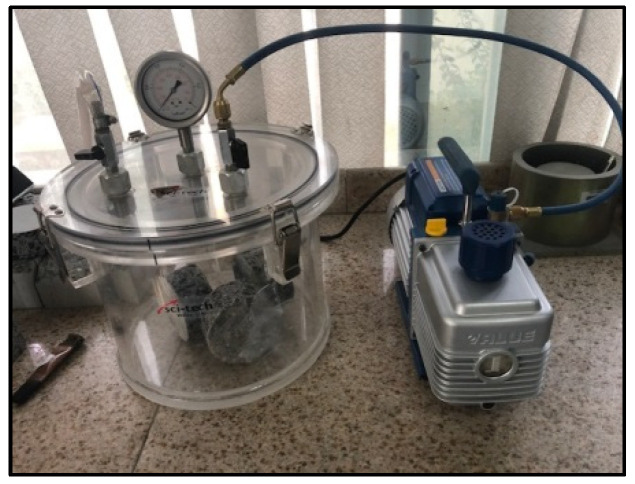
Water absorption test device.

**Figure 6 materials-15-07991-f006:**
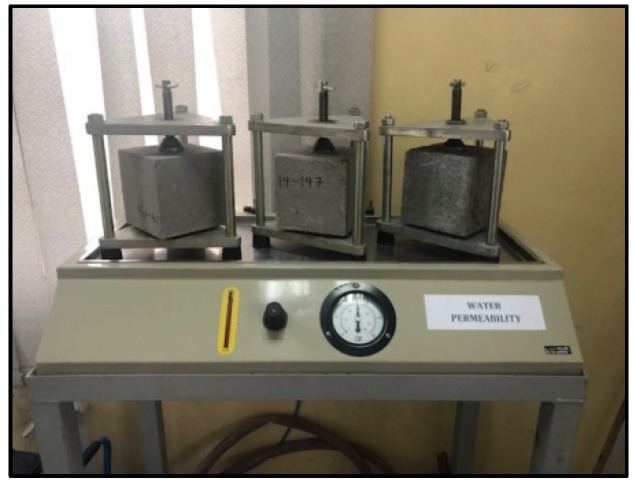
Water permeability test device.

**Figure 7 materials-15-07991-f007:**
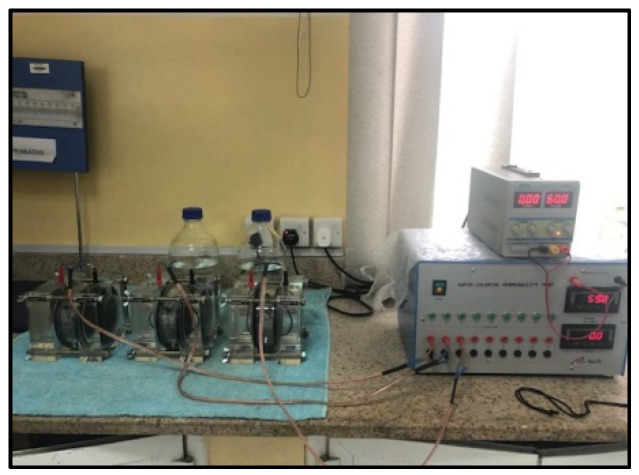
RCPT device.

**Figure 8 materials-15-07991-f008:**
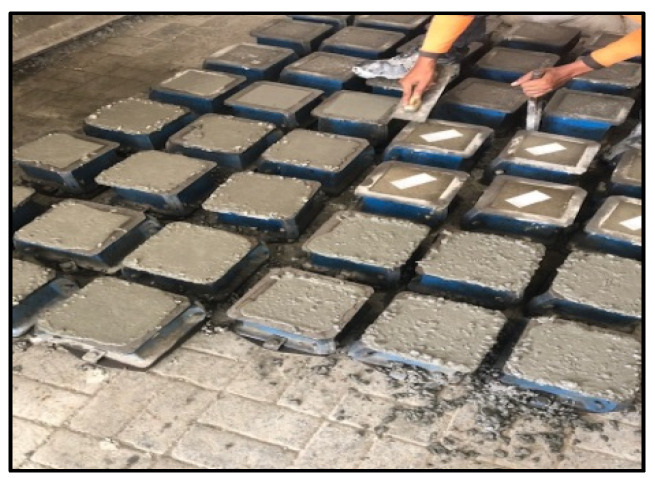
Sample preparation.

**Figure 9 materials-15-07991-f009:**
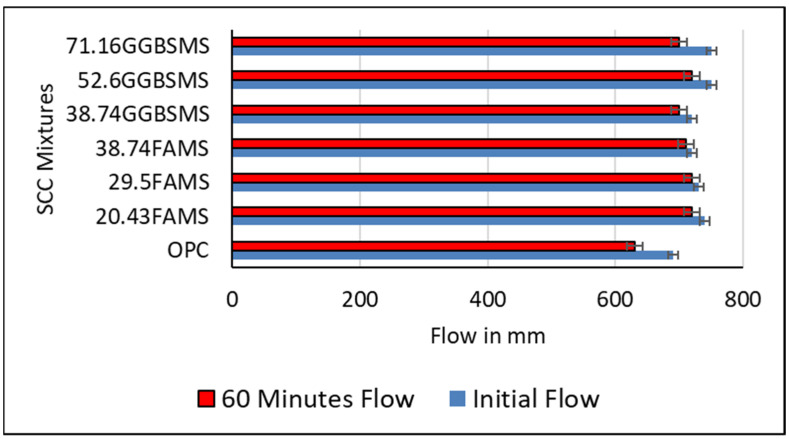
Slump-flow results for SCC.

**Figure 10 materials-15-07991-f010:**
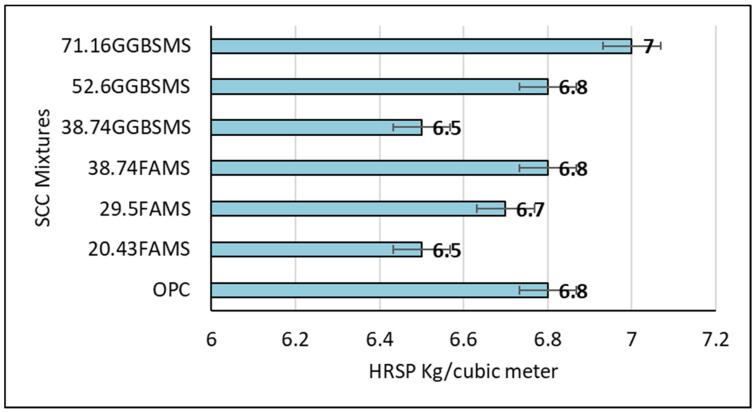
HRSP quantity used in SCC.

**Figure 11 materials-15-07991-f011:**
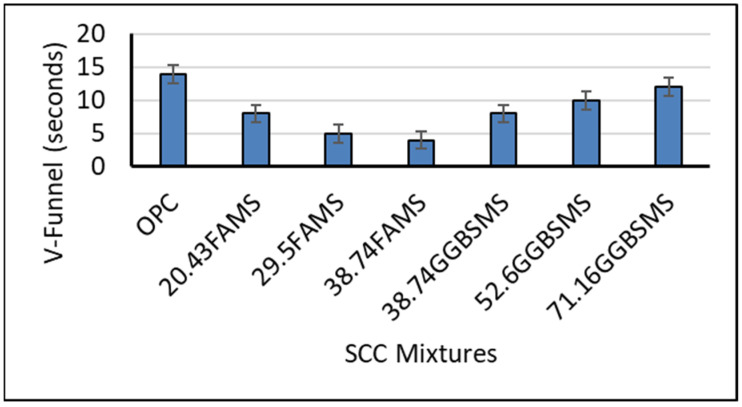
V-funnel results of SCC mixtures.

**Figure 12 materials-15-07991-f012:**
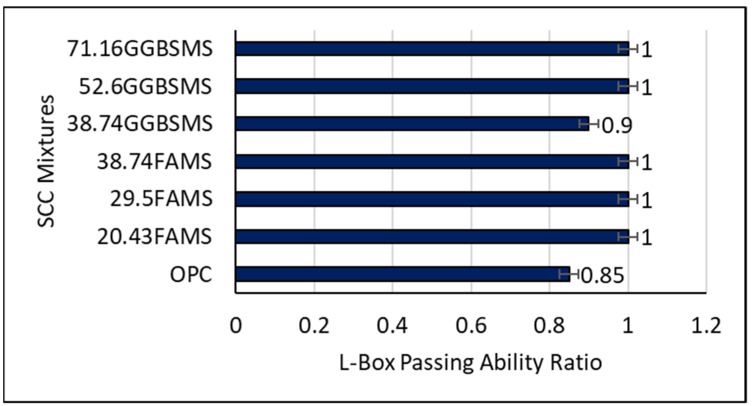
L-box test results (passing ability ratio).

**Figure 13 materials-15-07991-f013:**
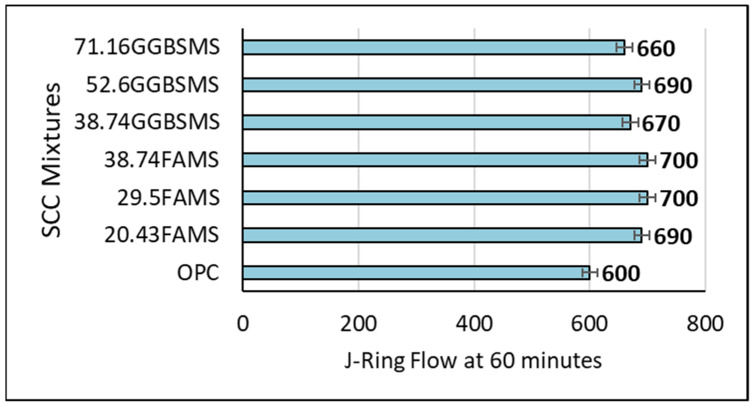
J-ring flow mm of SCC mixture.

**Figure 14 materials-15-07991-f014:**
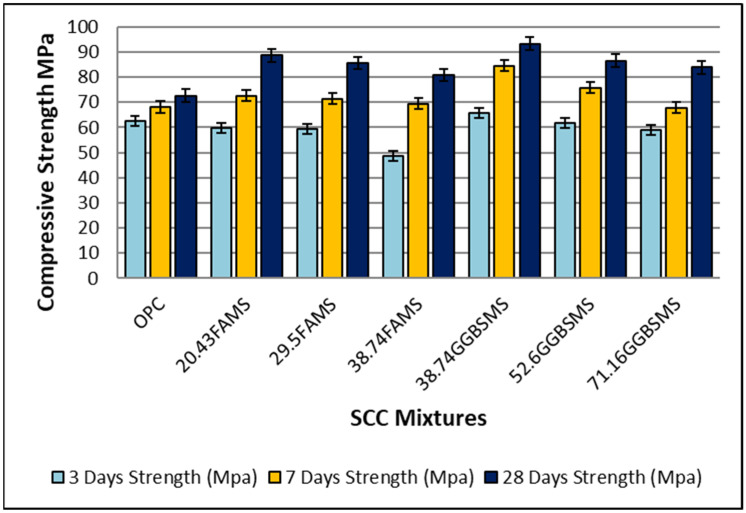
Compressive strength test results.

**Figure 15 materials-15-07991-f015:**
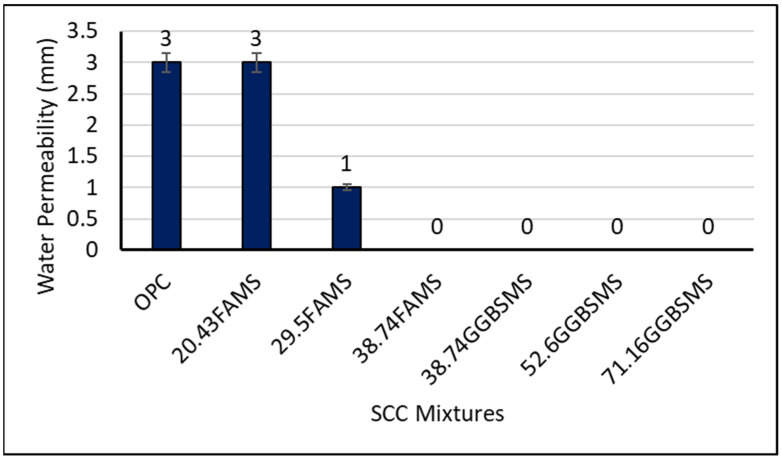
Water penetration depth of SCC mixtures at 28 days.

**Figure 16 materials-15-07991-f016:**
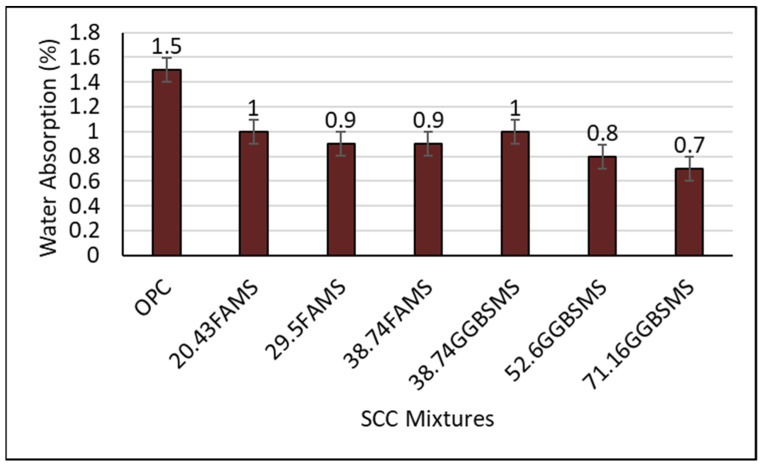
Water absorption of SCC mixtures at 28 days.

**Figure 17 materials-15-07991-f017:**
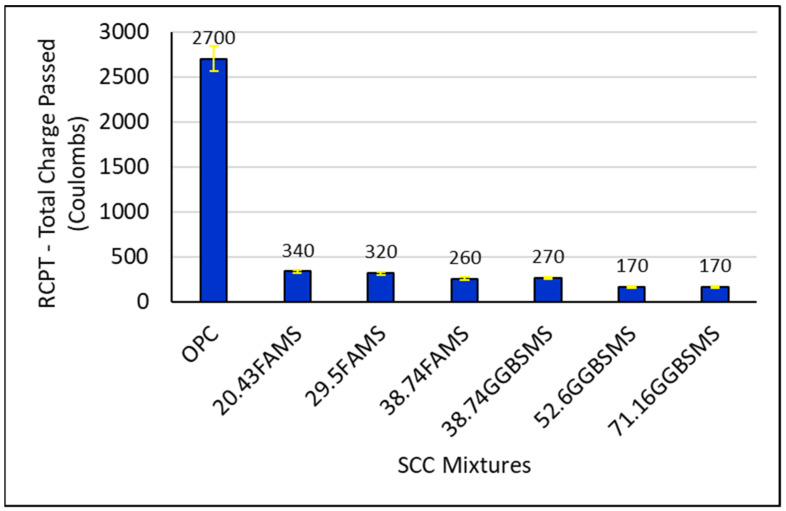
Total charge passed (RCP) of SCC mixtures at 28 days.

**Table 1 materials-15-07991-t001:** OPC physical properties.

Physical Properties	Specification	Results
Fineness (Air Permeability)	-	3280 cm^2^/gm
Initial setting time	≥60	205 Minutes
Final setting time		265 Minutes
Compressive Strength—2 Days	≥10	25.05 MPa
Compressive Strength—7 Days	-	40.22 MPa
Compressive Strength—28 Days	≥42.50 and ≤62.50	53.77 MPa

**Table 2 materials-15-07991-t002:** OPC chemical analysis.

Parameter	C_3_S	C_2_S	C_3_A	C_4_AF
Results	57.37%	13.83%	6.82%	11.32%

**Table 3 materials-15-07991-t003:** Chemical composition for cementitious materials.

Chemical Composition %	SiO_2_	Al_2_O_3_	Fe_2_O_3_	CaO	MgO	TiO_2_	SO_3_	Cl	Na_2_O	K_2_O	L.O.I
GGBS	31.27	13.34	0.64	41.55	6.90	0.98	0.11	0.01	-	-	-
FA	47.78	29.74	5.2	5.57	3.20	1.99	0.63	-	0.97	0.96	2.42
MS	92.38	-	-	-	-	-	-	-	0.46	-	5.01

**Table 4 materials-15-07991-t004:** Residue on 45 micron sieve for cementitious materials.

GGBS	1.78%
FA **[58]**	13.30%
MS	2%

**Table 5 materials-15-07991-t005:** SCC mixture proportions.

Mixture Code	Mixture without MS (%)	Mixture with MS (%)	OPC kg/m^3^	SCM kg/m^3^			Aggregate		Sand		HRSP kg/m^3^	Water kg/m^3^
				FA	GGBS	MS	20 mm	10 mm	Washed Sand	Dune Sand		
OPC	100% OPC	100% OPC	470	0	0	0	331	395	726	386	6.8	155
20.43FAMS	14.0% FA	20.43% (FA + MS)	374	66	0	30	338	401	711	374	6.5	155
29.5FAMS	23.4% FA	29.5% (FA + MS)	330	110	0	30	336	399	707	372	6.7	155
38.74FAMS	32.77% FA	38.74% (FA + MS)	286	154	0	30	335	397	704	370	6.8	155
38.74GGBSMS	32.77% GGBS	38.74% (GGBS + MS)	286	0	154	30	339	402	713	375	6.5	155
52.6GGBSMS	46.81% GGBS	52.6% (GGBS + MS)	220	0	220	30	338	401	711	374	6.8	155
71.16GGBSMS	65.5% GGBS	71.16% (GGBS + MS)	132	0	308	30	337	399	708	373	7.0	155

**Table 6 materials-15-07991-t006:** Fresh density for SCC mixtures.

Mixture	OPC	20.43% FAMS	29.5% FAMS	38.74% FAMS	38.74% GGBSMS	50.6% GGBSMS	71.16% GGBSMS
kg/m^3^	2490	2475	2480	2460	2485	2470	2470

## Data Availability

Data are available in a publicly accessible repository.

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
