# Peer review of "Strength and Durability of Sustainable Self-Consolidating Concrete with High Levels of Supplementary Cementitious Materials"

_materials, 2022, doi:10.3390/ma15227991_

Round 1
Reviewer 1 Report
The conducted work “Strength and Durability of Sustainable Self-Consolidating Concrete with High Levels of Supplementary Cementitious Materials” is good. However, following comments should be addressed to further improve paper:
A. GENERAL COMMENTS FOR PAPER ON OVERALL BASIS
1. Explicitly mention the novelty and research significance of current work in last paragraph of introduction section with emphasis on scientific soundness. Also, add recent relevant literature review from 2022 and 2021 papers in introduction section as there is no paper cited from 2022 and 2021.
2. Avoid long sentences throughout the manuscript, e.g. “Given all the advantages of SCC in practice, emphasis has been placed on optimizing its constituent composition. (e.g., supplementary cementitious materials (SCMs) and water-to-binder ratio), design (e.g., use of chemical admixtures such as superplasticizers and viscosity modifying admixture), and applications collectively affect SCC's fresh and hardened properties [12-14].” etc.
3. Avoid paragraph of few (1-3) sentences throughout the manuscript, particularly in results section, e.g. “On the other hand, in order to keep the water to binder ratio fixed at 0.33, high range water reducing and slump retention demand was gradually increased with the replacement of cement [59].”, etc.
4. Section 2 should have subheadings, like 2.1 Materials, 2.2 Concrete Preparations, and 2.3 Testing Procedures. In section 2.3, the sequence of testing procedures should be as per the sequence of results.
5. Results in current form look like a lab report. Results should be further elaborated with scientific reasoning.
6. A separate brief section (explaining the relevance of this research for practical implementation) may be added before conclusion section.
7. Conclusions are little long; these should be made brief and to the point. Closing remarks should be added at the end of conclusion section keeping in mind all conclusive bullet points.
8. English Language should be improved throughout the manuscript.
B. SPECIFIC COMMENTS FOR IMPROVING FOCUSSED RESEARCH
1. How the recommended SCC would be beneficial for application in industry?
2. The impact of different parameters should be collectively discussed for all possible combinations, e.g. compressive strength and water penetration of recommended SCC.
Author Response
Dear Reviewer,
Thanks for your comments and we already taken all the input. Really appreciate

Reviewer 2 Report
The paper is interesting but in this field there are numerous paper that replace this type of material with optimal results. From my point of view, there are assertions that should be complemented with electron microscopy or another type of test, since many of the defects that the specimens present internally are not visible to the naked eye, as is the case of water absorption. In the same way, there is a merger of regulations between European and American, which should be hominized to make the paper more credible, different regulations should be used when taking a reference, the test that is being carried out if is not contemplated, since it can generate misunderstanding in the investigation. On the other hand, I think that the study of waste materials for reuse is interesting and the more researchers who provide information about them, the better for their use. These would be my questions about the article in question:
1. What is HCA? When they are initialed, they must first comment on what they are referring to.
2. In the materials and method section, write GGB if you do not specify the type of slag or correct this abbreviation
3. Specify in the experimental program section which regulations have been used to carry out the tests and which one corresponds to each of them.
4. In results and discussion on the dosage of HRWR, it does not specify what it refers to.
5. The water absorption regulation BS 1881-122:2011+A1:2020 is the new one that is in force change the regulation
Author Response

(The authors gave the same response as above.)

Reviewer 3 Report
- “The production process of Portland cement releases at least 930 kg/ton of carbon dioxide into the atmosphere [17],…”, it is questionable. Below 850 kg CO2/ton cement clinker are generated in most of cement plant all over the world, the corresponding CO2 emissions/ton cement will be lower than 850 kg.
- The difference between micro silica used in this study and silica fume?
- Give the full name of HRWD when it is firstly appeared.
- Many more Figures.
- Results and discussions, three parts are adequate, such as “3.1 workability”, “3.2 mechanical performances” and “3.3 durability”.
- The combination of Fig.1 to Fig.8, the combination of Fig.9 to Fig.13 and the combination of Fig. 15 to Fig.17 are suggested. Meanwhile, the corresponding analysis and discussion should also be summarized.
- Conclusion should be brief.
Many more similar works have been reported, however, experiments and tests in this manuscript are sufficient, it can be accepted after minor revision.
Author Response

(The authors gave the same response as above.)

Round 2
Reviewer 2 Report
The authors have answered all the required questions